# On the Trade-off between Adversarial and Backdoor Robustness

**Cheng-Hsin Weng**     **Yan-Ting Lee**     **Shan-Hung Wu**
Department of Computer Science, National Tsing-Hua University, Taiwan, R.O.C.
{zxweng,ytlee}@datalab.cs.nthu.edu.tw, shwu@cs.nthu.edu.tw

## Abstract

Deep neural networks are shown to be susceptible to both adversarial attacks and backdoor attacks. Although many defenses against an individual type of the above attacks have been proposed, the interactions between the vulnerabilities of a network to both types of attacks have not been carefully investigated yet. In this paper, we conduct experiments to study whether adversarial robustness and backdoor robustness can affect each other and find a trade-off—by increasing the robustness of a network to adversarial examples, the network becomes more vulnerable to backdoor attacks. We then investigate the cause and show how such a trade-off can be exploited for either good or bad purposes. Our findings suggest that future research on defense should take both adversarial and backdoor attacks into account when designing algorithms or robustness measures to avoid pitfalls and a false sense of security.

## 1   Introduction

Deep neural networks (DNNs) have achieved impressive performance in many domains such as computer vision, natural language processing, speech, and robotics, etc. However, DNNs are shown to be susceptible to both adversarial attacks [14, 34] and backdoor attacks [7, 16, 25]. Adversarial attacks aim at fooling a model using examples (which are called adversarial examples) that are nearly indistinguishable from regular examples in human eyes or some distance measures in the input space. An adversarial example can be generated by slightly perturbing the input of a regular example in directions where the output of the model gives the highest loss. On the other hand, backdoor attacks aim at fooling the model with pre-mediated inputs. An attacker can "poison" training data by adding crafted triggers in some data points of a specific label. So, a model trained with poisoned data will perform well on a benign test set but behaves wrongly when the triggers are present in test data. The vulnerabilities of DNNs to these attacks raise concern about the robustness of security-critical machine learning applications, such as autonomous cars and speech recognition authorization.

Many defenses against adversarial or backdoor attacks have been proposed. In particular, the certified robustness [2, 3, 8, 10–13, 15, 21, 28, 35, 40–43] and adversarial training [14, 17, 19, 20, 24, 26, 29, 32, 44] are theoretically grounded and empirically strong methods, respectively, for defending adversarial attacks. To defend backdoor attacks, efforts have been made to detect and remove poisoned data (before training) [6, 36] or to fine-tune the model (after training) to unlearn backdoors [30, 39].

However, most existing defense methods are designed for one type of attacks only. The interactions between the vulnerabilities of a network to adversarial and backdoor attacks have not been carefully investigated yet. In practice, a model may be trained using the data collected from the public. It may also be deployed in an open environment where the input at runtime is accessible to the third party. As attackers could manipulate both training and testing data, it is crucial to understand how the interactions, if existing, will impact the current defenses.

In this paper, we conduct experiments to study whether the adversarial and backdoor robustness has an influence on each other. The answer is yes as we find a trade-off—by increasing the robustness of a network to adversarial examples via adversarial training, the network becomes more vulnerable to backdoor attacks. This finding is consistent on all the real-world datasets, including MNIST [23], CIFAR-10 [22], and ImageNet [9], and across all the settings we have tested. The trade-off delivers an important message: studying and defending one type of attacks at a time is dangerous because it may lead to a false sense of security. To elaborate this, we further show that new, subtle backdoor attacks can be created by exploiting the trade-off and that some well-known backdoor defenses [6,36] are *not* applicable to an adversarially robust model.

The trade-off is not entirely detrimental to existing defenses. We found that it conversely enhances a classes of backdoor defenses [30,39] that let a model unlearn backdoors after training. The following summarizes our contributions:

- We find that the adversarial robustness of a DNN is at odds with the backdoor robustness.
- We show, by conducting extensive experiments, that such a trade-off holds across various settings, including attack/defense methods, model architectures, datasets, etc.
- We investigate the reasons behind the trade-off by visualizing what is learned by the network.
- We demonstrate how an adversary can exploit the trade-off to create more concealed backdoor attacks and to make some existing backdoor defenses infeasible. Conversely, the trade-off also strengthens some other defenses.

Our findings have implications for both existing and future research. In particular, they give a guide on how to combine existing adversarial and backdoor defenses to achieve adversarial and backdoor robustness simultaneously. In addition, they open a door for joint adversarial and backdoor attack/defense in the future.

## 2    Related Works

**Adversarial attack and defense.** Studies [14,34] show that DNNs are vulnerable to adversarial examples. Based on different hypotheses about the cause of adversarial examples, a plethora of defense techniques against adversarial attacks has been proposed. Many of these methods, however, have been shown to fail [1,4,5]. The adversarial training [14,26] is one of the few surviving approaches and has shown to work well under many conditions empirically. Many recent defenses [17,19,20,24,29,32,44] are designed to work with or to improve adversarial training. Another major stream of defenses is the certified robustness [2,3,8,12,21,35], which provides theoretical bounds of adversarial robustness. Recent efforts [10,11,13,15,28,40–43] have been made to scale the certified robustness to larger networks and/or datasets at the cost of loose bounds.

**Backdoor attack and defense.** Studies [7,16, 25] show that a model can be injected backdoors (or "trojans") if it is trained by poisoned data, where some examples of a specific class contain crafted triggers, and behaves wrongly when the triggers are present in test data (see Figure 1). Depending on whether a trigger changes the label of an example or not, existing backdoor attacks can be divided into the dirty-label ones [7,16,25] and clean-label ones [31,38,45]. Generally, the clean-label attacks are preferred by adversaries because the poisoned examples have correct labels and therefore are harder to detect during data preprocessing or by a defender. However, a trigger of clean-label attacks, which is added to the input only, needs to be stronger (i.e., more learnable) to bias the model. Currently, most existing clean-label attacks either assume the model is pertained [31,45] or use an auxiliary model [38] to understand what features will be learned by the model and then use

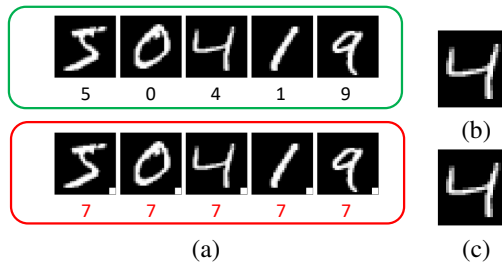

Figure 1: An example dirty-label backdoor attack on the MNIST dataset. (a) Benign (green) and poisoned (red) training data. (b) A test data instance without trigger will be correctly classified by the model. (c) A test instance with trigger (at the bottom-right corner) will always be predicted as the target label "7."

these features to enhance the trigger. The trigger enhancements in [38] look similar to adversarial perturbations, but they are static during the training phase and do not improve adversarial robustness.

Some techniques have been proposed to defend backdoor attacks, which can be roughly divided into the pre-training and post-training defenses. The pre-training approaches [6, 36] detect and remove poisoned data so a model can be properly trained. On the other hand, the post-training defenses [30, 39] reverse-engineers the potential triggers from a model with backdoors and then fine-tunes the model using a newly created dataset where the potential triggers are applied to data points of all classes. So, during the fine-tuning, the model will find the triggers useless for making correct predictions and thus unlearn backdoors.

**Interactions.** There are relatively few studies that take both adversarial and backdoor attack/defense into account. The study [27] shows that data poisoning can also be used to degrade the adversarial robustness of a model. Another study [33] uses backdoors as a honeypot that lures adversarial attacks into generating easy-to-detect adversarial examples. However, none of the above works studies a fundamental question: does the adversarial and backdoor robustness of a network affect each other?

## 3 The Trade-off and Its Cause

In this section, we show that simultaneous adversarial and backdoor robustness cannot be trivially achieved because there exists a trade-off between them. We also investigate the cause of the trade-off.

### 3.1 Experiments

As the adversarial training is currently commonly used to defend adversarial attacks, it is important to understand whether it affects backdoor robustness. To begin with, we train two networks of the same architecture using regular and adversarial training, respectively, and compare the backdoor robustness of the two models after training. We run the experiment on the MNIST, CIFAR-10, and ImageNet datasets.

**Settings.** We follow the settings used by [26] to configure the networks and training algorithms. Specifically, we use the projected gradient descent (PGD) with an $l_\infty$-norm constraint as the attack model of the adversarial training algorithm and set its parameters epsilon ($\epsilon$)/step size/number of iterations to 0.3/0.05/10 for MNIST, 8/2/5 for CIFAR-10, and 8/2/5 for ImageNet, respectively. In terms of network architecture, we use a naive CNN for MNIST, ResNet-32 for CIFAR-10, and pretrained ResNet-50 for ImageNet. We implement all the models using TensorFlow and train them on a cluster of machines with 80 NVIDIA Tesla V100 GPUs.[1]

**Evaluation.** To measure the backdoor robustness of the two networks, we devise a new clean-label backdoor attack, whose triggers are shown in Figure 2. We use different triggers for different datasets. The sizes of triggers are set to $3 \times 3$ pixels for MNIST and CIFAR-10, and $21 \times 21$ pixels for ImageNet. Note that this attack is weaker than the state-of-the-art clean-label backdoor attacks [31, 38, 45] because it does not use the weights of a pretrained or auxiliary model to "enhance" a trigger (i.e., to make the trigger easier to learn). However, the lack of the enhancement process prevents our backdoor

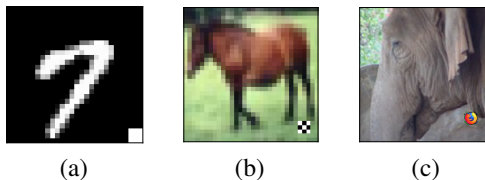

(a)  (b)  (c)

Figure 2: The backdoor triggers for (a) MNIST, (b) CIFAR-10, and (c) ImageNet used by our weak clean-label backdoor attack.

attack from interfering in the adversarial training. We randomly poison 5% of training examples by adding a backdoor trigger at the bottom-right corner of each poisoned image. The poisoned examples are sampled from the target class without label modification. After training the two networks using poisoned data, we evaluate the performance of the two networks by 1) (clean) accuracy on the benign test set, 2) adversarial robustness, that is, the accuracy on an adversarial test set generated by PGD, 3) success rate of the backdoor attack, which records the proportion of the poisoned test examples that are wrongly predicted as the target label by a model to all poisoned test examples.

Table 1: The trade-off between adversarial and backdoor robustness given different defenses against adversarial attacks. (a) Adversarial training and its enhancements. (b) Certified robustness.

| Dataset | Adv. Defense | Accuracy | Adv. Robustness | Backdoor Success Rate |
|---|---|---|---|---|
| MNIST | None (Std. Training) | 99.1% | 0% | 17.2% |
| | Adv. Training | 98.8% | 93.4% | 67.2% |
| | Lipschitz Reg. | 99.3% | 0% | 5.7% |
| | Lipschitz Reg. + Adv. Training | 98.7% | 93.6% | 52.1% |
| | Denoising Layer | 96.9% | 0% | 9.6% |
| | Denoising Layer + Adv. Training | 98.3% | 90.6% | 20.8% |
| CIFAR10 | None | 90% | 0% | 64.1% |
| | Adv. Training | 79.3% | 48.9% | 99.9% |
| | Lipschitz Reg. | 88.2% | 0% | 75.6% |
| | Lipschitz Reg. + Adv. Training | 79.3% | 48.5% | 99.5% |
| | Denoising Layer | 90.8% | 0% | 99.6% |
| | Denoising Layer + Adv. Training | 79.4% | 49% | 100% |
| ImageNet | None | 72.4% | 0.1% | 3.9% |
| | Adv. Training | 55.5% | 18.4% | 65.4% |
| | Denoising Layers | 71.9% | 0.1% | 6.9% |
| | Denoising Layers + Adv. Training | 55.6% | 18.1% | 68% |

(a)

| Dataset | Poisoned Data Rate | Adv. Defense | Accuracy | Certified Robustness | Adv. Robustness | Backdoor Succ. Rate |
|---|---|---|---|---|---|---|
| MNIST | 5% | None | 99.4% | N/A | 0% | 36.3% |
| | | IBP | 97.5% | 84.1% | 94.6% | 92.4% |
| CIFAR10 | 5% | None | 87.9% | N/A | 0% | 99.9% |
| | | IBP | 47.7% | 24% | 35.3% | 100% |
| | 0.5% | None | 88.7% | N/A | 0% | 81.8% |
| | | IBP | 50.8% | 25.8% | 35.7% | 100% |

(b)

**Results.** Table 1(a) shows the results of our experiment. By comparing the rows of standard and adversarial training, we can see that although the adversarial training improves adversarial robustness, it also degrades backdoor robustness—the success rate of the backdoor attack increases on all the MNIST, CIFAR-10, and ImageNet datasets. Specifically, our weak backdoor attack achieves more than 50% success rates on all datasets when applied to the adversarially trained network. This raises concern about the security of existing adversarially trained models. If their training data can be manipulated by an attacker, the models will have a high chance to predict whatever input as the target label set by the adversary.

**Does the trade-off hold for other adversarial defenses?** We implement two additional adversarial defenses, namely the Lipschitz regularization [17] and feature denoising layers [44], and test their performance.[2] The results, which are shown in Table 1(a) as well, show that 1) these two defenses do not work well when applied standalone, and 2) when paired up with the adversarial training, they results in the same trade-off.

We also examine the performance of the certified robustness defense IBP [15]. Although being relatively more scalable than other certified robustness defenses, the IBP still cannot scale to very deep networks and large datasets, including the ResNets and ImageNet dataset we used. So, we only conduct the experiments on the MNIST and CIFAR-10 datasets using the network and settings ($\epsilon = 0.4$ on MNIST) described in the original paper [15]. The results are shown in Table 1(b). Note

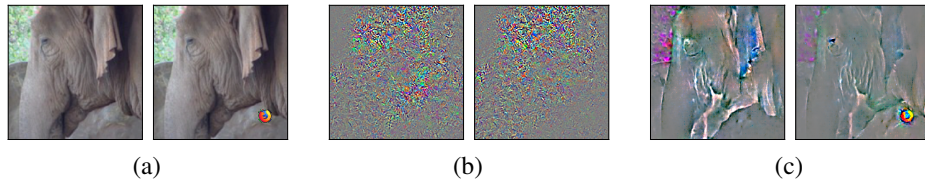

| (a) | (b) | (c) |

Figure 3: The saliency maps of the regularly and adversarially trained networks. (a) Benign (left) and poisoned (right) images from the ImageNet dataset. (b) Saliency maps of the regularly trained network given the benign (left) and poisoned (right) images. (c) Saliency maps of the adversarially trained network given the benign (left) and poisoned (right) images.

that the backdoor success rates saturate on CIFAR-10 when the poisoned data rate is 5%. Therefore, we reduce the poisoning rate to 0.5%. The results confirm the existence of the trade-off as well.

We further experiment with different settings, including the attack strengths and types, tolerance measures of adversarial perturbations, and model capacities. We also study additional backdoor triggers and their effects in Section 4. All the results reveal the same trend. For more details, please refer to Section 1 of the supplementary materials.

## 3.2 The Cause

To understand why an adversarially robust model is more vulnerable to backdoor attacks, we investigate what was learned by the model using visualization techniques. Figure 3 shows the saliency maps (i.e., the gradients of a model prediction with respect to the input) of the regularly and adversarially trained networks given a benign and poisoned image from the ImageNet, respectively. We can see that the adversarially trained network relies more on high-level features, which better align with human perception, to make a prediction. This is consistent with previous findings [18,37] that adversarial examples can be partially attributed to the presence of non-robust features (i.e., features that are highly predictive, yet brittle and incomprehensible to humans) in real-world datasets.

As an adversarially strong network relies more on robust, high-level features to make predictions, it also tends to learn from a backdoor trigger because the trigger provides robust features that are made to be strongly correlated with the target label. This explains the widespread existence of the trade-off we have discovered.

Note, however, that the above does not suggest that the high-level triggers are the only way to inject backdoors into an adversarially robust network. As we will see in Section 4, the channel triggers based on non-robust high-frequency pixel-level features can still successfully inject backdoors.

# 4 Exploiting the Trade-off

We show in this section how the trade-off discovered in Section 3 can be exploited for either good or bad purposes.

## 4.1 More Concealed Backdoor Triggers

A backdoor attacker needs to successfully place triggers in the training set in order to inject backdoors into a model. The triggers, however, may be detected and removed by humans or algorithms during data preprocessing. Here we study how the trade-off can be used to create backdoor triggers that are more subtle for humans to perceive. We will discuss how to evade the detection algorithms in the next subsection.

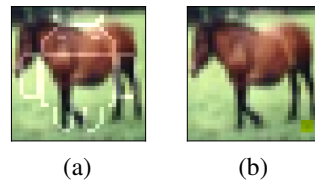

| (a) | (b) |

Figure 4: Example clean-label backdoor triggers of different types: (a) watermark and (b) channel. The channel trigger is added in the same position as the sticker trigger shown in Figure 2(b).

The backdoor attack used in Section 3 is a clean-label attack, which is already harder to detect than the dirty-labels ones [7,16,25]. In addition to using clean labels, we show that an adversary can make a trigger more subtle by

Table 2: The success rates of clean-label backdoor attacks given different (a) trigger types, (b) trigger sizes, (c) rates of poisoned data with the sticker triggers, and (d) trigger positions.

| Dataset | Adv. Defense | Trigger Type | Succ. Rate | Trigger Size | Succ. Rate | Poisoned Data | Succ. Rate | Trigger Pos. | Succ. Rate |
|---|---|---|---|---|---|---|---|---|---|
| MNIST | None | Watermark | 17.7% | $2 \times 2$ | 15% | 2.5% | 11.4% | Fixed | 17.2% |
| | Adv. Training | | **84.9%** | | **62.5%** | | **58%** | | **67.2%** |
| | None | Channel | N/A | $1 \times 1$ | 12.2% | 1% | 8.4% | Random | 4.6% |
| | Adv. Training | | N/A | | **57%** | | **52.6%** | | **59.9%** |
| CIFAR10 | None | Watermark | 84.2% | $2 \times 2$ | 47.1% | 2.5% | 30.8% | Fixed | 64.1% |
| | Adv. Training | | **90.9%** | | **99.9%** | | **95.4%** | | **99.9%** |
| | None | Channel | 33.5% | $1 \times 1$ | 31.1% | 1% | 15.2% | Random | 31.4% |
| | Adv. Training | | **72.4%** | | **69.8%** | | **88.9%** | | **95.1%** |
| ImageNet | None | Watermark | 13.4% | $14 \times 14$ | 3.2% | 2.5% | 1.6% | Fixed | 3.9% |
| | Adv. Training | | **46.8%** | | **49.6%** | | **46.6%** | | **65.4%** |
| | None | Channel | 1.1% | $7 \times 7$ | 3.7% | 1% | 0.6% | Random | 3.4% |
| | Adv. Training | | **16.4%** | | **18.2%** | | **20.8%** | | **63.5%** |
| | | (a) | | (b) | | (c) | | (d) | |

adjusting 1) trigger type, 2) trigger size, 3) poisoned data rate, and 4) trigger position, to be discussed later. We apply all the trigger variants to both the regularly and adversarially trained networks using the same settings described in Section 3 and then evaluate the performance of the networks. We find that the adversarial robustness of the two networks does not vary much given different triggers. Therefore, we focus on backdoor robustness. Note that all these trigger variants are of clean labels, and none of them has been reported to work without additional feature engineering [31, 38, 45] in the literature.

**Trigger type.** In addition to the "sticker" trigger type (see Figure 2), we consider two new types, namely the "watermark" and "channel" triggers, as shown in Figure 4. A channel trigger zeros out the blue channel of the pixels in a specific region. Table 2(a) shows the backdoor robustness of the two networks trained with these triggers.[3] By exploiting the trade-off, all types of triggers can be used to inject backdoors into the adversarially trained network. Note that the channel triggers, despite being non-robust from the human perspective, can still benefit from the trade-off to become a threat.

**Trigger size.** We also study how small can a trigger be to create a valid backdoor attack. We reduce the trigger sizes to $2 \times 2$ and $1 \times 1$ on MNIST and CIFAR-10, and to $14 \times 14$ and $7 \times 7$ on ImageNet, respectively. The results, which are summarized in Table 2(b), show that tiny triggers can still successfully inject backdoors into the adversarially trained network. Surprisingly, even the smallest possible triggers of size $1 \times 1$ achieve above 50% success rates on MNIST and CIFAR-10.

**Poisoned data rate.** Next, we see if an adversary can inject backdoors into the models when fewer examples are accessible. We reduce the poisoned data rates to 2.5% and 1% and use the sticker triggers to attack the two networks. As Table 2(c) shows, the trade-off greatly improves the efficiency of the backdoor attack against the adversarially trained network.

**Trigger position.** Finally, we study whether it is possible to create a backdoor attack by using the triggers that are placed at random corners of the training images. Table 2(d) shows the results. The attack works nicely against the adversarially trained network regardless of the trigger positions.

The clean-label, veiled backdoor attacks above, which are shown to work for the first time, motivate us to examine the effectiveness of existing backdoor defenses.

## 4.2 Making the Pre-Training Backdoor Defenses Infeasible

We consider three well-known backdoor defenses [6, 36, 39] and, to our surprise, find that two of them [6, 36] fail to defend the clean-label attacks in an adversarially trained network. Note that this

Table 3: The performance of the (a)(b) pre-training backdoor defenses [6,36] that detect and remove poisoned training data, and (c) post-training backdoor defense [39] that cleanses neurons.

| Dataset | Backdoor Attack | Detection Rate | | | Detection Rate | | |
|---|---|---|---|---|---|---|---|
| | | PR=5% | 1% | 0.5% | 5% | 1% | 0.5% |
| CIFAR10 | Dirty-Label Sticker + Std. Training | 81.6% | 24.4% | 2.4% | 100% | 100% | 5.58% |
| | Clean-Label Sticker + Adv. Training | 50.1% | 10.6% | 5.2% | 48.2% | 9.59% | 5.01% |
| ImageNet | Dirty-Label Sticker + Std. Training | 100% | 84.6% | 100% | 100% | 100% | 100% |
| | Clean-Label Sticker + Adv. Training | 50.5% | 13.1% | 9.23% | 47.8% | 9.67% | 3.72% |

(a)      (b)

| Dataset | Trigger Type | Trigger Label | Training Algorithm | Succ. Rate w/o Defense | Succ. Rate w/ Defense |
|---|---|---|---|---|---|
| CIFAR10 | Sticker | Dirty | Std. Training | 100% | 0.1% |
| | | Clean | Adv. Training | 99.9% | 0% |
| | Complex Watermark | Dirty | Std. Training | 99.7% | 39.3% |
| | | Clean | Adv. Training | 92.7% | 1.2% |
| ImageNet | Sticker | Dirty | Std. Training | 98.1% | 2.3% |
| | | Clean | Adv. Training | 65.4% | 1.1% |
| | Complex Watermark | Dirty | Std. Training | 96.3% | 39.8% |
| | | Clean | Adv. Training | 49.7% | 4.0% |

(c)

does not imply the two defenses are broken. They are just not applicable to an adversarially robust network.

The two works are both pre-training defenses (see Section 2 for the categorization of existing backdoor defenses) whose goal is to detect and remove poisoned data before training. A network with backdoors behaves normally when taking benign examples as input but wrongly predicts instances with triggers as the target label. This implies that there may be some neurons in the network that are activated by trigger features while others become active when normal features are present. Therefore, the distributions of neuron activations may be different for benign and poisoned examples. This enables the detection of poisoned examples by examining activations.

**Spectral signatures.** The work [36] first trains an auxiliary network using all examples, possibly with triggers, in a dataset. By assuming that the target label used by an attacker is known (which favors a defense), the defense 1) computes a vector of neuron activations in a hidden layer of the auxiliary network for each example of the target label, 2) extracts a spectral signature from each activation vector via the independent

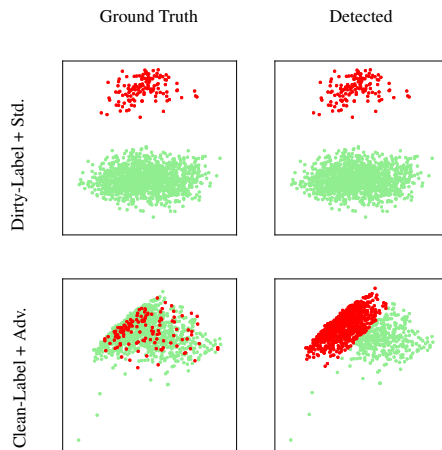

Figure 5: Distributions of benign (green) and poisoned (red) examples of the target label from ImageNet in the 2D-projected (using ICA) latent spaces of different models with backdoors.

component analysis (ICA), and 3) identifies the examples whose spectral signatures deviate most from the "center" (i.e., the average of all spectral signatures) as poisoned examples. Then, the final model can be trained by data excluding the detected examples. We implement the defense by using the activations of the third and fourth convolutional blocks to compute the spectral signature for each example on CIFAR-10 and ImageNet, respectively. Following the settings in Section 3, we apply it to

two networks where one is regularly trained by data with the dirty-label backdoor triggers [16] and another is adversarially trained by data with the clean-label attack shown in Figure 1. However, we found that, although improving the backdoor robustness of the regularly trained network, the defense does not significantly improve the backdoor robustness of the adversarially trained network against our newly proposed clean-label attack. See Section 2.1 of the supplementary file for more details. This is because of a degraded detection rate, i.e., the proportion of identified poisoned examples to all removed examples. Table 3(a) shows the detection rates of the defense given different poisoned data rates (PRs). The drop in the detection rates indicates that our newly proposed clean-label triggers are harder to detect. Taking advantage of the improved efficiency shown in Table 2(c), the non-detected triggers can still successfully inject backdoors into the adversarially trained network.

**Activation clustering.** The work [6] shares a similar idea to the above except it does not use the spectral signatures of neuron activations to detect triggers. Instead, it clusters examples into two groups based on a similarity measure in an activation space and then removes the group that is found suspicious by humans or algorithms. We implement this defense by removing the group that has more poisoned examples in the ground truth. After adversarially training a network using the pruned training set, we see little improvement in the backdoor robustness but a high cost of degraded performance for clean data. See Section 2.2 of the supplementary file for more details. Again, the poor detection rate is the cause, as shown in 3(b). Figure 5 shows the clusters found by the defense given a poisoned data rate of 1%. The poisoned data with clean-label triggers look very similar to the begin examples and can easily evade detection. In addition, there are many false positives, which decrease the number of benign training examples in the target class, creating data imbalance.

### 4.3 Enhancing the Post-Training Backdoor Defenses

Conversely, we found that the trade-off strengthens the defenses based on the neural cleansing [39], as shown in Table 3(c). The neural cleansing reverse-engineers potential triggers from a model with backdoors and then fine-tunes the model using the data containing potential triggers paired up with random labels to teach the model to unlearn backdoors (and cleanse neurons). Interestingly, it is currently considered to be weak for regularly trained models because it cannot reverse-engineer complex triggers [30]. While Table 3(c) confirms this (see the rows with complex, dirty-label triggers and regularly trained models), it also shows that the defense becomes strong when applied to the adversarially trained network. Figure 6 shows the complex backdoor trigger we used, and how the defense can successfully reverse-engineer it (at

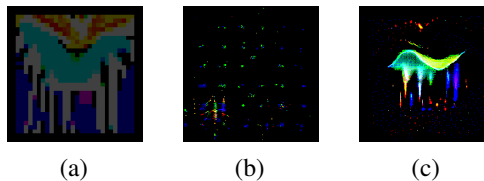

(a)　　　　(b)　　　　(c)

Figure 6: Reverse-engineered backdoor triggers on ImageNet. (a) Original complex watermark trigger used to poison training data. (b) Trigger reverse-engineered by [39] from the regularly trained network under the dirty-label backdoor attack. (c) Reverse-engineered trigger from the adversarially trained network under the clean-label backdoor attack.

least to a certain extent) from the adversarially trained network without using advanced enhancement techniques such as [30]. For now, pairing up adversarial training with neural cleansing (or its variants) seems to be a quick way to achieve adversarial and backdoor robustness simultaneously. The above also opens a door for joint adversarial and backdoor defense in the future.

## 5　Conclusion

In this paper, we showed, by using extensive experiments, the widespread existence of the trade-off between the adversarial and backdoor robustness of a DNN. We investigated the cause of the trade-off and demonstrated how an adversary can exploit it to create more concealed backdoor attacks and to make some pre-training backdoor defenses infeasible. Conversely, the trade-off strengthens some post-training backdoor defenses, which sheds light on joint adversarial and backdoor defense. As our future work, we plan to study algorithms and measures for joint adversarial and backdoor attack/defense. We will also study how the trade-off impacts real-world applications, such as self-driving cars, where the security of a machine learning system is in high demand.

## Broader Impact

Currently, the adversarial learning communities are aware of the adversarial attacks, backdoor attacks, and their respective defenses. However, the interactions between the vulnerabilities of a network to both types of attacks have not been carefully investigated yet. Our findings in this paper have implications for both existing systems and future research in adversarial learning. **The Bad.** The trade-off between the adversarial and backdoor robustness could be exploited by an adversary to create stronger and/or sneak attacks against existing security-critical machine learning systems and applications. Future research on defense should take both adversarial and backdoor attacks into account when designing algorithms or robustness measures to avoid pitfalls and a false sense of security. **The Good.** On the other hand, our findings give a guide on the selection of existing adversarial and backdoor defenses to achieve simultaneous adversarial and backdoor robustness. In addition, our finding that the trade-off improves the post-training backdoor defenses based on neural cleansing [39] also opens a door for joint adversarial and backdoor defense in the future. In particular, the "adversarial complement" of the work [30] on reverse-engineering triggers via generative distribution modeling seems to be a promising direction.

## Footnotes

[1]Our code is available at .

[2]We did not run Lipschitz regularization [17] on ImageNet because its memory requirement does not scale to 1000 classes.

[3]We do not apply the channel triggers to MNIST images because the images have only one channel.

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
