[Supplementary Material]

# On the Trade-off between Adversarial and Backdoor Robustness: Supplementary Materials

Cheng-Hsin Weng        Yan-Ting Lee        Shan-Hung Wu
Department of Computer Science, National Tsing-Hua University, Taiwan, R.O.C.
{zxweng,ytlee}@datalab.cs.nthu.edu.tw, shwu@cs.nthu.edu.tw

This document gives details that are omitted in the main paper due to space limitation. Specifically, Section 1 shows that the trade-off holds under various conditions, and Section 2 gives more experimental results of the pre-training backdoor defenses. Also, the code for the experiments is available at `https://github.com/nthu-datalab/On.the.Trade-off.between.Adversarial.and.Backdoor.Robustness/settings`. The model weights trained by different datasets can be found at

- MNIST: `https://drive.google.com/file/d/1F1ykVvmqZ9gNhIPRtx3qWgFv-RcX9j8a/view?usp=sharing`

- CIFAR-10: `https://drive.google.com/file/d/1OOY4eTPeSKmRgs_rscMDhTSLVpi0DiN4/view?usp=sharing`

- ImageNet: `https://drive.google.com/file/d/1M3DRe6O7kgIC02RWCQRwrTznbVYVmWyR/view?usp=sharing`

## 1 The Trade-off under Various Conditions

### 1.1 Attack Models and Strengths

An attack model is used by both the adversarial training and evaluation of adversarial robustness. To see whether the trade-off holds for different attack models, we experiment with different PGD settings as well as the FGSM attack [2]. In the former case, we adjust either the number of descent iterations or epsilon (i.e., the maximum allowable amount of perturbations) of PGD when generating an adversarial example. Table 1 summarizes the results, which show that the trade-off holds in spite of different settings of the PGD attack. Note that the second and following rows for each dataset should not be compared to each other because increasing epsilon or the number of iterations does not necessarily lead to a more adversarially robust model [5]. They should only be compared to the first row corresponding to the regular training. In the latter case, we replace the PGD with the FGSM attack. To avoid the label leaking problem [4], we use the targeted FGSM. Table 2 shows the results, and we can still see the existence of the trade-off.

### 1.2 Tolerance Measures of Adversarial Perturbations

Next, we study whether using different $p$-norms (that is, the tolerance measures of adversarial perturbations) in the adversarial training and evaluation will invalidate the trade-off. We run a new set of experiments using the $l_2$-norm and summarize the results in Table 3. Again, the trade-off holds in spite of different tolerance measures of adversarial perturbations.

### 1.3 Model Capacities

Finally, we test whether the capacity of a model has an influence on our finding. We run a new set of experiments by doubling the number of filters in each layer of a network. Table 4 shows the results, and we

Table 1: The trade-off holds across different PGD settings.

| Dataset | Epsilon | #Iter. | Adv. Defense | Accuracy | Adv. Robustness | Backdoor Succ. Rate |
|---|---|---|---|---|---|---|
| MNIST | N/A | N/A | None (Std. Training) | 99.1% | 0% | 17.2% |
| | 0.15 | 10 | Adv. Training | 99.3% | 94.8% | 37.7% |
| | 0.3 | 10 | Adv. Training | 93.4% | 93.4% | 67.2% |
| | 0.3 | 20 | Adv. Training | 94.7% | 94.7% | 57.7% |
| CIFAR10 | N/A | N/A | None | 90% | 0% | 64.1% |
| | 8 | 5 | Adv. Training | 79.3% | 48.9% | 99.9% |
| | 8 | 10 | Adv. Training | 76.5% | 43.8% | 100% |
| | 16 | 10 | Adv. Training | 62.8% | 31.4% | 100% |
| ImageNet | N/A | N/A | None (Std. Training) | 72.4% | 0.1% | 3.9% |
| | 8 | 5 | Adv. Training | 55.5% | 18.4% | 65.4% |
| | 8 | 10 | Adv. Training | 53.2% | 14.0% | 72.1% |
| | 16 | 10 | Adv. Training | 50.3% | 7.4% | 70.2% |

Table 2: The trade-of holds when the FGSM attack is used by the adversarial training and the evaluation of adversarial robustness.

| Dataset | Adv. Defense | Accuracy | FGSM Adv. Robustness | Backdoor Success Rate |
|---|---|---|---|---|
| MNIST | None (Std. Training) | 99.1% | 4.1% | 17.2% |
| | FGSM Adv. Training | 98.8% | 98.5% | 44.7% |
| CIFAR10 | None (Std. Training) | 90% | 22.1% | 64.1% |
| | FGSM Adv. Training | 84.9% | 65.1% | 83.7% |
| ImageNet | None (Std. Training) | 72.4% | 11.3% | 3.9% |
| | FGSM Adv. Training | 65.2% | 52.3% | 18.8% |

Table 3: The trade-off holds across different tolerance measures of adversarial perturbations.

| Dataset | $p$-Norm | Adv. Defense | Accuracy | Adv. Robustness | Backdoor Success Rate |
|---|---|---|---|---|---|
| CIFAR10 | $l_\infty$ | None (Std. Training) | 90% | 0% | 64.1% |
| | | Adv. Training | 79.3% | 48.9% | 99.9% |
| | $l_2$ | None (Std. Training) | 90% | 0.4% | 64.1% |
| | | Adv. Training | 79.7% | 48.3% | 99.9% |
| ImageNet | $l_\infty$ | None (Std. Training) | 72.4% | 0.1% | 3.9% |
| | | Adv. Training | 55.5% | 18.4% | 65.4% |
| | $l_2$ | None (Std. Training) | 72.4% | 0.7% | 3.9% |
| | | Adv. Training | 61.3% | 23.1% | 54.1% |

Table 4: The trade-off holds regardless of model capacities.

| Dataset | Model Architecture | Adv. Defense | Accuracy | Adv. Robustness | Backdoor Success Rate |
|---|---|---|---|---|---|
| CIFAR10 | [16,16,32,64] | None (Std. Training) | 90% | 0% | 64.1% |
| | | Adv. Training | 79.3% | 48.9% | 99.9% |
| | [32,32,64,128] | None (Std. Training) | 91.5% | 0% | 52.6% |
| | | Adv. Training | 83.7% | 50.4% | 99.8% |
| ImageNet | [64,128,256,512] | None (Std. Training) | 72.4% | 0.1% | 3.9% |
| | | Adv. Training | 55.5% | 18.4% | 65.4% |
| | [128,256,512,1024] | None (Std. Training) | 71.1% | 0.7% | 16.8% |
| | | Adv. Training | 57.0% | 20.6% | 68.5% |

can see that that model capacity does not seem to affect the trade-off.

The above results, together with the results shown in the main paper, demonstrate the widespread existence of the trade-off between adversarial and backdoor robustness.

## 2    Pre-Training Backdoor Defenses

Here, we provide more experimental results to support the claims in Section 4.2 of the main paper.

### 2.1    Spectral Signatures

Following Section 4.2 of the main paper, we conduct experiments to evaluate the performance of the pre-training defense based on spectral signatures [6]. We apply the defense to the two networks, where one is regularly trained by data with the dirty-label backdoor triggers [3] and another is adversarially trained by data with the newly proposed clean-label attack. Table 5(a) shows the results. As we can see, the backdoor robustness of the adversarially trained network is not significantly improved by the defense, although the mean deviation of the spectral signatures of remaining examples from the center does reduce greatly. This is due to a low detection rate—the defense can only detect about 50% of the poisoned examples with the clean-label triggers. As shown in Table 3(a) of the main paper, the detection rate becomes even lower when the poisoned data rate decreases. This leaves many non-detected triggers in the pruned training set, and they can still successfully inject backdoors into the adversarially trained network.

### 2.2    Activation Clustering

Table 5(b) shows the performance of the pre-training defense based on activation clustering [1]. As we can see, the defense does not significantly improve the backdoor robustness of the adversarially trained network on CIFAR-10. Again, this is because of a low detection rate, which drops below 50% as compared to the 100% given by the regularly trained network under the dirty-label attack.

On ImageNet, the backdoor robustness of the adversarially trained network seems to be improved as the success rate of the backdoor attack is reduced from 65.4% to 11.9%. However, the detection rate remains low (47.8%), and we speculate that the improvement is due to the fact that the defense removes too many (benign) examples of the target label, therefore making the training data imbalanced. In this case, the trained model will predict other classes more often than the target label, which increases backdoor robustness but degrades the recall of benign test data of the target class. To verify our hypothesis, we test and observe a recall of 8% for the target class given by the adversarially trained model with the defense, which is much

Table 5: The success rates of the backdoor attacks against the pre-training defenses based on (a) spectral signatures [6] and (b) activation clustering [1].

| Dataset | Backdoor Attack | Succ. Rate w/o Defense | Succ. Rate w/ Defense | Detection Rate | Deviation |
|---------|-----------------|------------------------|-----------------------|----------------|-----------|
| CIFAR10 | Dirty-Label Sticker + Std. Training | 100% | 98.9% | 81.6% | 16.7 |
|         | Clean-Label Sticker + Adv. Training | 99.9% | 97.1% | 50.1% | 0.08 |
| ImageNet | Dirty-Label Sticker + Std. Training | 98.1% | 0.1% | 100% | 151.7 |
|          | Clean-Label Sticker + Adv. Training | 65.4% | 58.7% | 50.5% | 2.39 |

(a)

| Dataset | Backdoor Attack | Succ. Rate w/o Defense | Succ. Rate w/ Defense | Detection Rate |
|---------|-----------------|------------------------|-----------------------|----------------|
| CIFAR-10 | Dirty-Label Sticker + Std. Training | 100% | 0.7% | 100% |
|          | Clean-Label Sticker + Adv. Training | 99.9% | 97.5% | 48.2% |
| ImageNet | Dirty-Label Sticker + Std. Training | 98.1% | 0.1% | 100% |
|          | Clean-Label Sticker + Adv. Training | 65.4% | 11.9% | 47.8% |

(b)

lower than the 38% recall given by the model without the defense. In summary, the backdoor robustness is improved at the cost of degraded performance for clean data.