[Reviews · NeurIPS 2020]

Review 1

Summary and Contributions: In this paper, the authors conducted an empirical analysis to demonstrate the existence of a trade-off between the vulnerability of deep neural networks to adversarial examples and backdoor attacks. They show that defenses used to mitigate the impact of adversarial examples tend to increase vulnerability to backdoor attacks, and propose novel backdoor attacks to bypass (some of the) existing defenses.

Strengths: + Identification and empirical investigation of a novel trade-off between vulnerability to adversarial examples and backdoor attacks.

Weaknesses: - The trade-off is investigated only empirically and with qualitative explanations - Lack of a proper discussion on threat modeling - Unclear categorization of defenses to backdoor attacks

Correctness: Yes, the claims are mostly correct, as well as the empirical methodology

Clarity: Yes, to some extent. A more structured discussion on threat models and backdoor defenses is lacking.

Relation to Prior Work: Not really. Apart from the identification of the aforementioned trade-off, it is not clear how the proposed work builds on the literature of backdoor defenses and attacks. It is also unclear in which applications and practical cases it makes sense to consider both the threat of adversarial examples and backdoor attacks – and how one should build defenses or mitigate these threats when this happens.

Reproducibility: Yes

Additional Feedback: I discuss below the three main concerns that I have with this paper. 1. Lack of a more in-depth analysis of the identified trade-off. This work makes an interesting point on highlighting a trade-off between adversarial vulnerability to adversarial examples and backdoor attacks. While the empirical evaluation confirms this trade-off on the reported cases, the qualitative explanations provided in terms of the dependency of classifiers on robust or non-robust features seem to be not grounded on neither theoretical nor empirical solid evidences. In this respect, I think that the paper would be much more valuable if the authors could try to categorize the trade-off in a more formal way, identifying the key factors contributing to it. For instance, previous work by Simon-Gabriel et al., 2019, has identified the size of input gradients as a factor that impacts on vulnerability to adversarial examples. Then, does reducing it (i.e., learning smoother functions via proper regularizers or adversarial training) opens up vulnerability to backdoor attacks? How/Why? - C.-J. Simon-Gabriel, Y. Ollivier, L. Bottou, B. Sch ̈olkopf, and D. Lopez-Paz. First-order adversarial vulnerability of neural networks and input dimension. In K. Chaudhuri and R. Salakhutdinov, editors, 36th ICML, volume 97 of PMLR, pages 5809–5817, 2019. 2. Threat model. The authors claim in both the abstract and the introduction that “Our findings suggest that future research on defense should take both adversarial and backdoor attacks into account when designing algorithms or robustness measures to avoid pitfalls and a false sense of security.” First of all, the sentence ‘false sense of security’ was used by Athalye et al. [1] to denote failures of existing defenses against adversarial examples which were not properly evaluated. By properly evaluated, I mean under the same threat model (e.g., against an L2-norm attack bounded by a certain eps). In this work, instead, the authors implicitly use this sentence but changing the underlying threat model used in the original defense papers. I think this is not completely fair. It is obvious that a defense which was designed to work against a given attack is not necessarily robust (or even more vulnerable) to another. This also happened in defenses against adversarial examples when changing the attack norm; e.g., from L2 to L1. I would recommend the authors to clarify this point – i.e., that they are considering a different threat model w.r.t. the original papers, and to also report practical cases in which considering both such threats against the same model can be considered pertinent/relevant. 3. Unclear categorization of backdoor defenses (and attacks). Finally, I feel that there is a bit of confusion in the description of backdoor defenses. In fact, many such defenses have different goals. Neural cleanse aims to detected if the model is backdoored, while Activation clustering and Spectral aim to detect whether a dataset is poisoned. Other defenses even try to find if a specific input is a backdoor. These details are not discussed, and defenses to backdoors are not properly categorized. Here too, the threat model for these defenses is not reported. Some of them make indeed specific assumptions on the trigger size and shape, while other do not. These details are however necessary to understand whether the newly-proposed attacks by the authors adhere to the same threat model of the original defense, or not. In the second case, in fact, I think it is not fair to say that the defense is broken. A defense is broken if, under the same threat model, you can find an attack that decreases its performance below the nominal one reported in the original paper. Otherwise you are just showing that with a completely different threat model the defense does not work (e.g., using a trigger different from the one assumed by the defense). *** Comments to the authors' rebuttal *** I thank the authors for their response, and for running an additional experiment to check the impact of input gradient regularization on adversarial robustness and on backdoor attacks. However, my comment was more on trying to better motivate why the trade-off identified by the authors exist, e.g., in relationship to model complexity/regularization, and moving away from a mere empirical analysis. I still find the motivation given in Sect. 3.2 too weak and not systematically investigated (it's only based on an image example). While I believe that the paper has potential and identifies an interesting phenomenon, I think its analysis of the root causes remains too preliminary and I would invite the authors to better characterize the trade-off in more formal terms (even in approximate cases, e.g., under linear approximations).


Review 2

Summary and Contributions: This paper shows the trade-off between the adversarial robustness and backdoor robustness. The experiments on MNIST, CIFAR10, ImageNet partly support the finding.

Strengths: The idea is novel. The experiments and analysis on MNIST, CIFAR10, ImageNet partially support the finding that there's a trade-off between adversarial and backdoor robustness. It is also tested on backdoor defenses.

Weaknesses: 1. The study among different adversarially trained models is missing, thus the trade-off is unclear among robust trained models. For example, the TRADES model may improve both the robustness and back-door robustness. 2. Following the point above, it is unclear whether the trade-off still holds when the models that are partially adversarial robust. Since the results are present in two extreme without the middle results. For example, models with 10%,20%, 30% adversarial robustness accuracy. A curve with some reasonable resolution is needed to show the trade-off. 3. Experiment details missing. It is unclear to the reviewer whether the data for the adversarial training is poisoned or not. Would adversarial training still work under poison data? Would that mean successful backdoor attack (weak back-door robustness) also reduce the adversarial robustness? Maybe a figure showing the trade-off under this setting is missing. 4. Too few steps of attack for adversairal attack (only 5 to 10 steps), it is may not access the true adversarial robustness.

Correctness: The number of adversarial attack step is too few (only 5 to 10 steps), the reviewer would recommend try PGD and CW with ~200 steps with several random starts. Too few steps give a false sense of robustness.

Clarity: Yes.

Relation to Prior Work: Yes.

Reproducibility: Yes

Additional Feedback: See weakness. I thank the authors' response and the experiment results. My concern is partly addressed. However, from the results we can see when the model gets robust, the model is more vulnerable under backdoor attacks. But Table 3 not showing the transition phase when the backdoor is partly success, Table 2 does not compare TRADES with other robust models. I also agree with other reviewers that the source of the trade-off need to be explored. Thus I raise my score to a 5, weak reject.


Review 3

Summary and Contributions: The paper studies the trade-off between adversarial robustness and robustness to backdoor attacks. The authors argue that models trained to be robust to adversarial perturbations are vulnerable to backdoor injections, allowing for very simple and innocuous attacks to be effective. This hypothesis is evaluated on a range of experimental settings. === POST-REBUTTAL UPDATE === I appreciate the authors' response and the additional experimental results provided. At this point, it is clear that the observed trade-off is real. At the same time, I do agree with the other reviewers that both the precise source of the trade-off, as well as the exact factors influencing it (e.g., degree of model robustness) can be explored further. I thus lower my score to a 6 (weak accept).

Strengths: The main idea of the paper is quite interesting, uncovering a somewhat subtle interaction between robust training and backdoor attacks. The intuition is quite clear in hindsight: the backdoor trigger is a robust and helpful feature that is even more likely to be picked up in the setting of robust training (where other features are rendered useless). The experimental evaluation is quite extensive, covering a wide range of scenarios.

Weaknesses: I found the terminology "breaking a defense" quite misleading. Traditionally, this phrase is used to indicate that the adversary is modifying the attack to circumvent a particular defense. However, in this case, the attacker is not really doing anything different, the defense simply stops working when the defender performs robust training. I believe that clarifying this point in the manuscript is crucial, since it creates confusion about the scope of the work. My score is conditional on the fact that the authors understand this issue and edit the manuscript appropriately.

Correctness: Based on my understanding, the experimental methodology is sound. It is somewhat unfortunate that the main results of the paper are obtained for an adversary that poisons 5% of the training set and thus, in the case of CIFAR10, 50% of the target class---a not very realistic scenario. At the same time, the authors do provide experiments for lower poisoning percentages, demonstrating that their conclusions are robust.

Clarity: The paper is well-written and easy to understand.

Relation to Prior Work: Prior work has been adequately discussed.

Reproducibility: Yes

Additional Feedback: L130: I interpret 5% of the dataset to correspond to 50% of the target class for CIFAR10, is this correct? How does this translate to ImagetNet (1000 classes)? The spectral signatures defense removes a fraction of training points based on their correlation with the top eigenvector. How is the fraction determined for different poisoning ratios? How does a "dirty" attack perform in the case of robust training? Similarly, better, or worse than standard training?

[Author Response · NeurIPS 2020]

To all reviewers who provided feedback, thank you. Based on the common comments about the writing, we will use your suggestions on how to improve on it. In particular, we will stop using the phrase "**breaking a backdoor defense**" to avoid confusion. Instead, we will conclude that "**existing backdoor defenses may not be applicable to an adversarially trained or robust network**." We would like to address your other concerns in the following:

**To Reviewer 1. Q1:** Does reducing the input gradients (Simon-Gabriel et al., 2019) opens up vulnerability to backdoor attacks? **A1:** Thanks for this good question. We implemented the work by adding a regularization term to the loss of a model. As we increase the strength of the regularization term, the adversarial robustness of the model increases as well on CIFAR-10 while the backdoor robustness decreases, as shown in Table 1. The input gradients give another way to explain the trade-off and we will add the results to our paper. **Q2:** Theoretically grounded experiments. **A2:** The certified robustness methods (see line 27) are theoretically grounded defense methods because they target the worst-case adversarial robustness. We showed in lines 151-157 that even a certified robustness method IBP (Sven Gowal et al., 2019) are subject to the trade-off. **Q3:**

| Table 1 | Input Gradient Regularization Strength | | | |
|---|---|---|---|---|
| | 0 | 0.005 | 0.01 | 0.035 |
| Adv. Robustness | 0 | 0.007 | 0.025 | 0.132 |
| Backdoor Succ. Rate | 0.453 | 0.802 | 0.889 | 0.993 |

What are practical cases that need to consider both adversarial and backdoor threats against the same model? **A3:** Any security-sensitive model that may be under attack *during data collection* and *after training* should consider both threats. Examples span from self-driving cars that learn from public scenes to face-detection systems built on an open collection of face images. **Q4:** Unclear categorization of backdoor defenses. **A4:** Thanks. We will improve our writings based on your suggestions and avoid using the phrase "breaking a backdoor defense." We hope the above clarifies your concerns and convinces you to improve your rating.

**To Reviewer 2. Q1:** The TRADES model may improve both the robustness and back-door robustness. **A1:** Thanks for this good question. We run TRADES on CIFAR-10 using the code and settings provided by the authors and found that the trade-off still holds, as shown in Table 2. **Q2:** It is unclear whether the trade-off still holds when the models that are partially adversarially robust. **A2:** This is an interesting direction to explore. We make a model "partially" adversarially robust by adversarially training it with a PGD attack

| Table 2 | TRADES | |
|---|---|---|
| | Reg. Trained | Adv. Trained |
| Adv. Robustness | 0 | 0.543 |
| Backdoor Succ. Rate | 0.275 | 0.988 |

that has a smaller $\epsilon$ (i.e., the maximum allowable perturbations to the input). This makes the model less robust when it is evaluated by a PGD attack with a larger $\epsilon$ at test time. Table 3 shows the results on CIFAR-10. As we can see, the trade-off still holds. In particular, the backdoor robustness of the model seems to degrade quickly as the adversarial robustness increases. **Q3:** Are the data for the adversarial training poisoned or not? **A3:** Yes. **Q4:** Would that mean successful backdoor attack also reduces adversarial robustness? **A4:** Possibly. But in practice, we observed very little difference in adversarial robustness across different models and datasets. **Q5:** Too few steps of attack for adversarial attack (only 5 to 10 steps), it may not access the true adversarial robustness. **A5:** Following your suggestion, we evaluate the adversarial robustness of the adversarially trained models using the PGD attack with 200 steps on MNIST and CIFAR-10. Table 4 shows the results, which indicate that the models have reasonable adversarial robustness. We hope the above clarifies your concerns and convinces you to improve your rating.

| Table 3 | Train-Time $\epsilon$ | | | |
|---|---|---|---|---|
| | 4/255 | 8/255 | 12/255 | 16/255 |
| Adv. Robust. ($\epsilon = 16/255$) | 0.119 | 0.257 | 0.306 | 0.312 |
| Backdoor Succ. Rate | 0.993 | 1 | 0.999 | 0.999 |

| Table 4 | PGD Steps | |
|---|---|---|
| | 5 | 200 |
| Adv. Robustness on MNIST | 0.93 | 0.92 |
| Adv. Robustness on CIFAR10 | 0.45 | 0.39 |

**To Reviewer 3. Q1:** The main idea of the paper is quite interesting, and the intuition of the trade-off is quite clear in hindsight. **A1:** Thank you for your positive comments. **Q2:** Line 130: I interpret 5% of the dataset to correspond to 50% of the target class for CIFAR10, is this correct? **A2:** Yes. We also experimented with 10% and 5% of the target class in Section 4.2. **Q3:** How does this translate to ImagetNet? **A3:** 0.05% of training data means 50% of the target class. **Q4:** In spectral signatures, ... how is the fraction of removed training data determined for different poisoning ratios? **A4:** Here we assume the defender knows the number of poisoned examples, so we remove the same amount of examples from training data. This favors the defense. **Q5:** Does a "dirty-label" backdoor attack perform better or worse in the case of robust training than in standard training? **A5:** Good question. Our results showed that the attack achieves similar backdoor success rates in the two cases. We hope the above clarifies your concerns and humbly ask that if you think our findings deserve attention to the field, please champion this paper.

[Meta-Review · NeurIPS 2020]

This paper is postulating a trade-off between adversarial robustness and resilience to backdoor attacks. The underlying phenomenon is definitely of interest. However, upon closer inspection, it was not clear if the way the authors phrase their findings is the most illuminating/appropriate one. (See the reviews for more details.) Specifically, what the authors seem to be finding (and also mentioning in the paper) is that robust models tend to rely on different features than non-robust models and, because of that, class-consistent data poisoning triggers tend to be picked up by robust models more. It seems that structuring the paper (and provided evidence) around this claim will lead to much less confusion (see the reviews and important points made in them). Overall, this is a paper that would be worth having in NeurIPS should the authors reflect the above point (as well as the ones made in the reviews).